# A Lateral Flow Immunoassay Coupled with a Spectrum-Based Reader for SARS-CoV-2 Neutralizing Antibody Detection

**DOI:** 10.3390/vaccines10020271

**Published:** 2022-02-10

**Authors:** Rui-Lin Huang, Yi-Chen Fu, Yung-Chih Wang, Chitsung Hong, Wei-Chieh Yang, I-Jen Wang, Jun-Ren Sun, Yunching Chen, Ching-Fen Shen, Chao-Min Cheng

**Affiliations:** 1Institute of Biomedical Engineering, National Tsing Hua University, Hsinchu 30013, Taiwan; h646132@gmail.com (R.-L.H.); sandy216621@gmail.com (Y.-C.F.); yunching@mx.nthu.edu.tw (Y.C.); 2National Defense Medical Center, Department of Internal Medicine, Division of Internal Medicine and Tropical Medicine, Tri-Service General Hospital, Taipei 11490, Taiwan; wystwyst@gmail.com; 3Spectrochip Inc., Hsinchu 302059, Taiwan; cthong23@gmail.com; 4Ping An Medical Clinic, Tainan 708, Taiwan; dr.albertyang@gmail.com; 5Department of Pediatrics, Taipei Hospital, Ministry of Health and Welfare, New Taipei City 24213, Taiwan; wij636@gmail.com; 6College of Public Health, China Medical University, Taichung 406040, Taiwan; 7School of Medicine, National Yang Ming Chiao Tung University, Taipei 11221, Taiwan; 8National Defense Medical Center, Institute of Preventive Medicine, Taipei 23742, Taiwan; tsghsun@gmail.com; 9Department of Pediatrics, National Cheng Kung University Hospital, College of Medicine, National Cheng Kung University, Tainan 70101, Taiwan; 10Institute of Clinical Medicine, College of Medicine, National Cheng Kung University, Tainan 70101, Taiwan

**Keywords:** coronavirus disease 2019 (COVID-19), lateral flow immunoassay, neutralizing antibody, severe acute respiratory syndrome coronavirus-2 (SARS-CoV-2), AstraZeneca

## Abstract

As of August 2021, there have been over 200 million confirmed case of coronavirus disease 2019 caused by severe acute respiratory syndrome coronavirus and more than 4 million COVID-19-related deaths globally. Although real-time polymerase chain reaction is considered to be the primary method of detection for SARS-CoV-2 infection, the use of serological assays for detecting COVID-19 antibodies has been shown to be effective in aiding with diagnosis, particularly in patients who have recovered from the disease and those in later stages of infection. Since it has a high detection rate and few limitations compared to conventional enzyme-linked immunosorbent assay protocols, we used a lateral flow immunoassay as our diagnostic tool of choice. Since lateral flow immunoassay results interpreted by the naked eye may lead to erroneous diagnoses, we developed an innovative, portable device with the capacity to capture a high-resolution reflectance spectrum as a means of promoting diagnostic accuracy. We combined this spectrum-based device with commercial lateral flow immunoassays to detect the neutralizing antibody in serum samples collected from 30 COVID-19-infected patients (26 mild cases and four severe cases). The results of our approach, lateral flow immunoassays coupled with a spectrum-based reader, demonstrated a 0.989 area under the ROC curve, 100% sensitivity, 95.7% positive predictive value, 87.5% specificity, and 100% negative predictive value. As a result, our approach exhibited great value for neutralizing antibody detection. In addition to the above tests, we also tested plasma samples from 16 AstraZeneca-vaccinated (ChAdOx1nCoV-19) patients and compared our approach and enzyme-linked immunosorbent assay results to see whether our approach could be applied to vaccinated patients. The results showed a high correlation between these two approaches, indicating that the lateral flow immunoassay coupled with a spectrum-based reader is a feasible approach for diagnosing the presence of a neutralizing antibody in both COVID-19-infected and vaccinated patients.

## 1. Introduction

The rapid spread of severe acute respiratory syndrome coronavirus 2 (SARS-CoV-2) reached pandemic proportions in December 2019 [1]. The infectious disease caused by this pathogen has been officially named “Coronavirus disease” (COVID-19) by the World Health Organization (WHO, Geneva, Switzerland). As of 30 September 2021, there have been more than 233,136,147 confirmed COVID-19 cases and 4,771,408 deaths [2]. Those affected by COVID-19 have been found to produce a neutralizing antibody (N_Ab_) against SARS-CoV-2 immediately [3]. N_Ab_ has also been detected in the convalescent plasma of COVID-19 patients and recovered patients [4,5]. The neutralizing antibody titer is usually measured using a plaque reduction neutralization test, focus reduction neutralization test, or a neutralizing assay [6]. Severely afflicted patients have higher N_Ab_ titers and longer titer existences than those with mild and asymptomatic infections [7]. The N_Ab_ level has been found to be associated with immune protection from symptomatic SARS-CoV-2 infection [8].

The development and implementation of vaccines has become the solution to ameliorating the pandemic. However, the emergence of various SARS-CoV-2 variants has diminished the protective capacity of the vaccination [9]. The effectiveness of these vaccines in the long term, therefore, becomes a rather important topic [8]. As the COVID-19 pandemic continues, reinfections with SARS-CoV-2 have been found in those with previous infections, as well as those who have been vaccinated [10]. Detection of the N_Ab_ levels may be an important means of stratifying the patient risk level and may be considered a critical element of public health decision-making.

A lateral flow immunoassay is a simple technique employing solid-phase immunoassay technology that combines the principles of thin-layer chromatography and immune recognition reaction. However, there are restrictions in sample volumes and the lack of an enhancing enzyme limit test sensitivity [11]. We have proposed the use of a quantitative spectrum-coupled lateral flow immunoassay to enhance the detection limit for the IgG antibody in COVID-19-infected patients [12]. In this study, we used a lateral flow immunoassay accompanied by a spectrum-based reader for detecting SARS-CoV-2 N_Ab_ in those with COVID-19, as well as those that received the AstraZeneca vaccine (ChAdOx1nCoV-19). The conventional ELISA protocols were also used for both comparison and validation.

## 2. Materials and Methods

### 2.1. Patients and Samples

Blood samples (serum used in the study) were obtained within 12 h of arriving in hospital from patients (age 18–74, mean 40, and median 34) diagnosed as COVID-19-positive from April 2020 to June 2021 at two hospitals in Taiwan: Tri-Service General Hospital and Taipei Hospital of Ministry of Health and Welfare in Northern Taiwan. Ten adults without fever, respiratory symptoms, or any sign of acute infection, as well as having never been vaccinated against COVID-19 and never been infected by SARS-CoV-2, were enrolled as healthy individuals. The study protocol was approved by the institutional review board of each site (TSGH IRB No. C202005067 and TH IRB No. TH-IRB-0020-0011). The diagnosis of COVID-19 was confirmed by positive real-time polymerase chain reaction test results for SARS-CoV-2 from nasopharyngeal samples. We also collected blood samples (EDTA plasma used in the study) and saliva samples from AstraZeneca (ChAdOx1nCoV-19)-vaccinated healthy individuals (age 23–69, mean 37, and median 29). These participants all received two doses of AstraZeneca vaccines 8–12 weeks apart at Taiwan’s National Cheng Kung University Hospital. Blood samples were taken one month after full vaccination, and saliva samples were collected 3–5 days after full vaccination. Participants provided written informed consent upon recruitment. The study protocol was approved by the institutional review board of National Cheng Kung University Hospital (NCKU IRB No. A-BR-110-051).

### 2.2. Reader & Reflectance Spectral Analysis

The spectrum analyzer (in collaboration with Taiwan SpectroChip Inc.; Taiwan FDA: MD (I)-008090 and US FDA: 3017810861) was equipped with a lateral flow immunoassay for detecting the SARS-CoV-2 N_Ab_ reflectance spectrum from this immunoassay. This device provided a continuous spectrum and captured the high-resolution reflectance spectrum of the immunoassay test line region via an optical module. The spectrum-based reader provided high-resolution (3–5 nm) results across a vast spectral range (300–1100 nm). The primary reflectance wavelengths detected using the spectrum reader were 500 nm and 600 nm, with a main reference wavelength of 680 nm. The reflectance results were log-transformed to absorbance via the formula Absorbance = −log(reflectance). ΔA*_Test/Control line_* was calculated as the difference between absorbance (at maximum, 500–600 nm) and absorbance (at 680 nm). The N_Ab_ value was calculated using the ratio of the ΔA*_Test line_* value to ΔA*_Control line_* value:ΔA*_Test line_* = Absorbance (at maximum 500 nm to 600 nm) − Absorbance (at 680 nm)
ΔA*_Control line_* = Absorbance (at maximum 500 nm to 600 nm) − Absorbance (at 680 nm)
N_Ab_ = 3 – ΔA_Test line_/ΔA_Control line_

In this formula, ΔA*_Test/Control line_* refers to the color reflection value of the optical scanning SARS-CoV-2 N_Ab_ lateral flow immunoassay. The lower the ΔA*_Test/Control line_* ratio, the lower the color intensity of the test line region, indicating that the sample contains a higher concentration of the neutralizing antibody. Hence, the result is reversed by adding a minus sign in front of ΔA*_Test/Control line_*. Furthermore, when the result is a negative number, a value of three is added to the value in order to analyze the data. Therefore, a higher value indicates a higher N_Ab_ concentration.

### 2.3. Lateral Flow Immunoassay (SARS-CoV-2 Neutralizing Antibody Rapid Test Cassette)

The SARS-CoV-2 Neutralizing Antibody Rapid Test Cassette (Healgen Scientific LLC, Houston, TX, USA) is a rapid test that utilizes a combination of spike protein receptor-binding domain antigen-coated gold particles for the detection of neutralizing antibodies to SARS-CoV-2 in human sera. This rapid test cassette is a lateral flow immunochromatographic assay based on the principle of competitive binding. When using this test, N_Ab_ that may be present in blood against the spike protein receptor-binding domain antigen conjugates for binding sites on the angiotensin-converting enzyme 2 receptor. When an adequate volume of test sample is dispensed into the sample well of the rapid test cassette, the sample migrates by capillary action along the cassette. N_Ab_ to SARS-CoV-2, if not present in the blood, will not saturate the binding sites of the spike protein receptor-binding domain antigen coated on the particles. The spike protein receptor-binding domain antigen-coated particles will then be captured by the angiotensin-converting enzyme 2 receptor, which is precoated in the test line region, and a visible red line shows up in the test line region. The color intensity of the test line decreases as the concentration of N_Ab_ increases. The red line does not form in the test line region if there are sufficient N_Ab_ in the blood, because they saturate all the binding sites of the spike protein receptor-binding domain antigen coated on the particles. To serve as a procedural control, a red line will always appear in the control line region (C in Figure 1), indicating that the proper sample volume is added and membrane wicking occurred. For both the serum and plasma samples, one drop (approximately 25 µL) is transferred to the sample well (S in Figure 1) of the rapid test cassette; one drop of buffer (approximately 40 µL) is then placed. After 10 min, the red line appears, and the rapid test cassette can be placed into the spectrum analyzer immediately to read the results (Figure 1).

### 2.4. Enzyme-Linked Immunosorbent Assay

The N_Ab_ of blood samples and saliva samples were determined using the surrogate virus neutralization test kit (SARS-CoV-2 Surrogate Virus Neutralization Test Kit, GenScript, Piscataway, NJ, USA). The result was performed according to the manufacturer’s instructions. A total of 100 µL of sample and controls were diluted 1:9 in the sample dilution buffer and then mixed with 100 µL of horseradish peroxidase-conjugated receptor-binding domain antigen. After the mixtures were incubated at 37 °C for 30 min, 100 µL of mixtures were transferred to a 96-well plate coated with recombinant protein of angiotensin-converting enzyme 2 receptor. After incubation at 37 °C for 15 min, the supernatant was removed, and the plate was washed four times with 260 µL of washing buffer each time. Then, 100 µL of tetramethylbenzidine substrate were added to the plate, which was incubated for 15 min at room temperature. A total of 50 µL of stop solution were added to stop the reaction. The 96-well plate was detected immediately by an ELISA reader (Molecular Devices, San Jose, CA, USA) at 450 nm. The percentage of inhibition (inhibition%) was calculated according to the following formula:(1)$$\mathrm{Inhibition}\%=(1-\frac{OD450\;value\;of\;sample}{average\;OD450\;value\;of\;negative\;control})\;\times 100\%$$

The SARS-CoV-2 Spike Protein IgG ELISA Kit (E-EL-E602; Elabscience, Houston, Texas, USA) was used to detect the SARS-CoV-2 spike protein IgG antibody in samples. First, a total of 100 µL of sample or control solution was added to the well of a 96-well microplate, which was incubated for 45 min at 37 °C. The solution was then aspirated from each well, and the wells were washed three times with 350 µL of washing buffer. A total of 100 µL of horseradish peroxidase-conjugated mouse anti-human IgG working solution was added to each well, and the plate was incubated for 30 min at 37 °C. After that, the washing process was repeated five times. A total of 90 µL of substrate reagent was added to each well, and the plate was incubated for about 15 min at 37 °C in the dark. Last, a total of 50 µL of stop solution was added to each well. The OD value was determined using an ELISA reader at 450 nm (Molecular Devices, San Jose, CA, USA). The cut-off was calculated as cut-off = 0.13 + negative control average A450 (when the negative control average A450 < 0.05, calculate it as 0.05; if 0.05 ≤ negative control average A450 ≤ 0.10, calculate it as the actual value). If the sample absorbance ≥ cut-off, the sample was classified as positive for the SARS-CoV-2 spike protein IgG antibody; if the sample absorbance < cut-off, the sample was classified as negative.

Human total IgG antibody was determined using the Human IgG ELISA Kit (ab195215; Abcam, Cambridge, UK). First, the samples were diluted 1:2 × 10^6^, and standards 1–8 were prepared with the concentrations as follows: 15, 7.5, 3.75, 1.87, 0.93, 0.47, 0.23, and 0 ng/mL. Then, the capture and detector antibodies were diluted in antibody diluent CP. A volumetric total of 50 µL of sample or a standard and antibody cocktail combined were added to wells of a precoated 96-well microplate. The plate was incubated for 40 min at room temperature on a plate shaker. Afterwards, each well was washed three times with 350 µL of washing buffer. Then, 100 µL of tetramethylbenzidine development solution were added to each well, and the plate was incubated for 5 min in the dark on a plate shaker. Last, a total of 100 µL of stop solution was added to each well, and the plate was shaken for one minute. The OD value was recorded at 450 nm according to the manufacturer’s instructions with an ELISA reader (Molecular Devices, San Jose, CA, USA), and the amount of human IgG was calculated by using a standard curve.

### 2.5. Statistical Analysis

The correlation between the two different methods, lateral flow immunoassay coupled with a spectrum-based reader and conventional ELISA, was measured using the Spearman’s rank correlation coefficient and the Bland-Altman plot. A *p*-value of less than 0.05 was considered to be statistically significant. The area under the receiver operating characteristic (ROC) curve (AUC) was used to evaluate the diagnostic ability of N_Ab_.

## 3. Results

### 3.1. COVID-19 Mild and Severe Cases

The workflow of a SARS-CoV-2 N_Ab_ rapid test cassette coupled with a spectrum-based reader is shown in Figure 1. The principle of the lateral flow immunoassay designed to detect SARS-CoV-2 N_Ab_ provided a visible, qualitative result (i.e., Yes or No) with a 10-min duration. Scanning the lateral flow immunoassay with a light and tiny reader facilitated the acquisition of a quantitative result. Blood samples from those with COVID-19 (26 mild cases and four severe cases), 10 healthy individuals, and 16 vaccinated patients were used to detect N_Ab_ using the lateral flow immunoassay coupled with a spectrum-based reader (Table 1 and Table 2). The presence of a band at the control line region (C) on all lateral flow immunoassays validated the tests for various N_Ab_ concentrations. The results were interpretated according to the manufacturer’s instructions, which defined the results as follows. Weak Positive—Two lines are visible, and the color intensity of the line in the test line region (T) is the same as the control line region. Middle Positive—Two lines are visible, and the color intensity of the line in the test line region (T) is weaker than the line in the control line region (C). High Positive—Only one line is visible on the control line region (C), and there is no visible line in the test line region (T). Negative—Two lines are visible, but the line in the test line region (T) is stronger than the line in control line region (C). We tested the different cases, as shown in Figure 2A,B, and it was difficult to interpret the color band intensity by the naked eye, in some cases. Therefore, we established a quantitative approach using a spectrum analyzer to analyze the color intensity. We found that the reflectance spectra of N_Ab_ were significantly different between the control line region and the test line region at around 540 nm (Figure 2C).

Due to the above results, we derived a mathematical formula for describing the color intensity between the control line region and test line region. The reflectance spectra were used to acquire a N_Ab_ value for constructing a N_Ab_ concentration. Since it was difficult to prepare a standard SARS-CoV-2 N_Ab_ substance, we pretested the correlation assessed with a series dilution of a severe case patient’s serum to measure the stability of the system. As shown in Figure 2(D1), the result was analyzed using the Hill equation, which indicated a very high correlation (R-squared = 0.9582) and extracted lower than 3% dilution data, as shown in Figure 2(D2) (R-squared = 0.9649). Based on the N_Ab_ concentrations acquired by our approach, COVID-19-infected and vaccinated patients tended to have higher N_Ab_ concentrations compared to the healthy controls, and the difference reached statistical significance (*p* < 0.0001) (Figure 2E).

We then measured the N_Ab_ concentrations with a dilution series of a severe case patient’s serum using two methods: (1) SARS-CoV-2 Neutralizing Antibody Rapid Test Cassette coupled with a spectrum-based reader and (2) ELISA kit. As shown in Figure 3A, the correlation between the N_Ab_ concentration results from lateral flow immunoassay approach, and the concentrations based on ELISA were highly relevant and statistically significant (Rho = 0.9818, *p*-value < 0.0001). Therefore, we tested all the samples, as shown in Figure 3B, and discovered a similar result (Rho = 0.9288, *p*-value < 0.0001). All of the results are shown in Table 1. Furthermore, to validate the agreement between these two methods, we employed a Bland–Altman analysis to evaluate any bias between the mean differences and to estimate an agreement interval (95%) between the lateral flow immunoassay approach and ELISA. The Bland–Altman plot simply represents every ratio between the two paired methods against the average of the measurement. As shown in Figure 3C, the mean ratio of the two methods was 30.33, and the limits of agreement (±1.96 standard deviation) were 0.8757 and 59.79. The width of the limits of agreement was 58.9143, which included most of the data points. The ROC curve is a common method for summarizing the performance of each classifier into a single measure. It shows a trade-off between sensitivity and specificity. We set the percent inhibition over 30 percent to be 1 for our at-state variable and less than 30 percent inhibition to be 0 for our at-state variable and then set the N_Ab_ value as our test variable. As shown in Figure 3D, the area under the curve (AUC) values for both the lateral flow immunoassay and ELISA were 0.989, *p* < 0.001. The N_Ab_ cut-off value according to the lateral flow immunoassay ROC curve was 1.005, the sensitivity was 100%, the specificity was 87.5%, the positive predictive value was 95.65%, and the negative predictive value was 100%.

### 3.2. Vaccinated Persons

A total of 16 AstraZeneca-vaccinated patients were enrolled to test their N_Ab_, once again, using two methods: (1) SARS-CoV-2 Neutralizing Antibody Rapid Test Cassette coupled with a spectrum-based reader and (2) the ELISA kit. The correlation between these two methods is shown in Figure 4A. The N_Ab_ value based on the lateral flow immunoassay was highly related to the percent inhibition based on ELISA, with Rho = 0.9412, and *p*-value < 0.0001. The results of the N_Ab_ analysis based on the lateral flow immunoassay and ELISA are shown in Table 2. A Bland–Altman analysis was used to assess the agreement between these two methods. Using the mean and the standard deviation of the differences between two measurements, we obtained the agreement interval (95%) of the lateral flow immunoassay and ELISA. The result is shown as a Bland–Altman plot in Figure 4B. The mean difference of the two methods was 18.47, and the limits of agreement, which are defined as the mean difference plus and minus 1.96 times the standard deviation of the differences, were 34.6 and 2.347. The interval of the limits of agreement contained almost all the data points. In Figure 4C, the ROC curve is presented as AUC = 0.933. The N_Ab_ cut-off value according to the lateral flow immunoassay ROC curve was 1.8926, the sensitivity was 80%, the specificity was 100%, the positive predictive value was 100%, and the negative predictive value was 75%. We also attempted to analyze the reproducibility of the same lateral flow immunoassay coupled with a spectrometer. The means of the Nab levels of the three vaccinated persons verified three times were 0.5935, 1.8368, and 1.3545, and their standard deviations were 0.0203, 0.0295, and 0.1341, showing the stable reproducibility of our lateral flow immunoassay coupled with a spectrum-based reader.

We also tested the SARS-CoV-2 spike protein IgG antibody and human total IgG antibody in patients who received two doses of the AstraZeneca vaccine to compare the N_Ab_ results. For the spike protein IgG antibody, from the OD value of the sample, the cut-off index was defined as the ratio of the assay signal to cut-off signal. It was predicted that the amount of spike protein IgG antibody had something to do with N_Ab_ [13], and the correlation analysis proved this result with a Rho = 0.5724 and *p*-value = 0.0035 (Figure 5A). From the spike protein IgG antibody results, we determined the correlation between the human total IgG antibody and N_Ab_ based on ELISA, as shown in Figure 5B, with a Rho = 0.0251 and *p*-value = 0.8794. Taken together, we concluded that there was no specific correlation. One reason for this may be that there were too many factors influencing the total human IgG antibody, so the response after vaccination was not significant enough.

In order to make testing N_Ab_ easier for everyone, we wanted to know if saliva could be used to examine the neutralizing antibody content. Spike protein IgG and IgA antibodies and receptor-binding domain protein IgG and IgA antibodies can be detected in saliva after two doses of the vaccine [14]. We consequently collected and examined saliva from 22 patients following their second vaccine dose. When testing these patients’ N_Ab_ using the sera samples, most showed positive results; nevertheless, when testing using saliva samples, almost all of them showed negative results (Figure 5C). In summary, the concentration of N_Ab_ in saliva is too low to be a detection tool.

## 4. Discussion

During the COVID-19 pandemic, hundreds of thousands of people have been infected with SARS-CoV-2 every day. It is still important to diagnose COVID-19 by different methods. Serological assays are meaningful for those who have recovered from the disease and in the late stage of infection. It can also provide information that can be used to assess the disease state among the population. COVID-19 may remain a health threat for years to come. As with influenza, it is critical to detect COVID-19 at the “front lines”, including such places as clinics, airport, and barracks. Among an array of potential serological test methods, lateral flow immunoassays are ideal for large-scale screening and point-of-care testing. However, the read-out is primarily qualitative (Yes or No) and is generally not considered sensitive enough for diagnosing serious infections, such as COVID-19. To improve the quantitative capacity of the lateral flow immunoassay, we integrated a newly designed spectrum analyzer into our SARS-CoV-2 Neutralizing Antibody Rapid Test Cassette procedures. We showed that the SARS-CoV-2 neutralizing antibody could be detected using a commercially available lateral flow immunoassay coupled with a portable spectrum device to provide point-of-care testing. Our results indicate that this newly quantitative system has high sensitivity and specificity.

In this study, we used a SARS-CoV-2 Neutralizing Antibody Rapid Test Cassette with a spectrum-based reader to find a N_Ab_ value to present the concentration of the neutralizing antibody in COVID-19-infected and vaccinated patients (Table 1 and Table 2). Compared with healthy controls, most COVID-19 patients and vaccinated people had higher neutralizing antibody concentrations, but there were some samples that had very low concentrations. There may be some other factors we need to discuss, for example, the number of days that patients were infected with SARS-CoV-2. The total days of infection may produce varying responses associated with early-stage, late-stage, or even very-late exhausted immune system states. We confirmed a high correlation between our lateral flow tool and conventional ELISA for determining the virus-neutralizing antibody level. Our results indicated the capacity to obtain quantitative data and demonstrated that a higher N_Ab_ value was equivalent to a higher neutralizing antibody concentration. We also tried to test the correlation between the IgG antibody (SARS-CoV-2 spike protein IgG antibody and total IgG antibody) and neutralizing antibody (Figure 5A,B). We found a correlation between the neutralizing antibody and SARS-CoV-2 spike protein IgG antibody but no correlation to the total IgG antibody. In addition to the serum or plasma, saliva was also tested, because it was easier to obtain. However, the content of the neutralizing antibody in saliva 3–5 days after the second vaccination dose was too low to be measured (Figure 5C). Testing saliva is still a promising approach, but testing samples days after the second dose of vaccine may provide better results. Nevertheless, there were still some additional limitations in this study: (1) The number of samples tested was low. (2) For vaccinated people, we only tested patients that received the AstraZeneca vaccine. (3) Among the COVID-19 patients sampled, there were none infected with variants such as the delta variant. We are continuing to propose efforts to perform research on COVID-19-vaccinated patients. In the near future, we will continue tracking COVID-19 patients and vaccinated patients to verify the sensitivity and specificity and to observe the vaccine efficacy using two methods that we demonstrated in this study at extended time points following vaccination. These tests will be used to verify the utility of our spectrum-based approach. We may also integrate an IL-6 test strip [15] into our system. This combination may be more practical. It could improve the diagnostic distinction between severe and mild COVID-19 cases by using two biomarkers while reducing the testing time and cost.

## 5. Conclusions

In summary, we used a lateral flow immunoassay coupled with a spectrum-based reader as an alternative approach to conventional ELISA to assess the presence and level of the COVID-19 neutralizing antibody in both afflicted and vaccinated patients. Although this lateral flow immunoassay approach demonstrated less specificity than ELISA, out results showed that the COVID-19 neutralizing antibody rapid test coupled with a spectrometer has very high sensitivity and specificity, which means our rapid test system has great diagnosis ability. Furthermore, compared with ELISA, rapid test savings in time, cost, and training were significant. Conventional ELISA requires completion by well-trained individuals in a laboratory setting, but the lateral flow immunoassay can be operated by anyone after reading the instructions and can even be completed by individuals in a home setting. So far, SARS-CoV-2 has continuously mutated and infected people around the world. Although many people have been vaccinated against COVID-19, according to the research, the function of the vaccine will decrease with time, and the falling levels are associated with individual differences. Hence, we think that it is necessary to detect neutralizing antibodies in the home. We can monitor our own protection at any time. Moreover, there is no need to go to a high-risk field, such as a clinic or hospital, for testing the neutralizing antibodies. At the same time, the data collected from our spectrometer can analyze big data to evaluate whether or not to get a vaccine booster.

## Figures and Tables

**Figure 1 vaccines-10-00271-f001:**
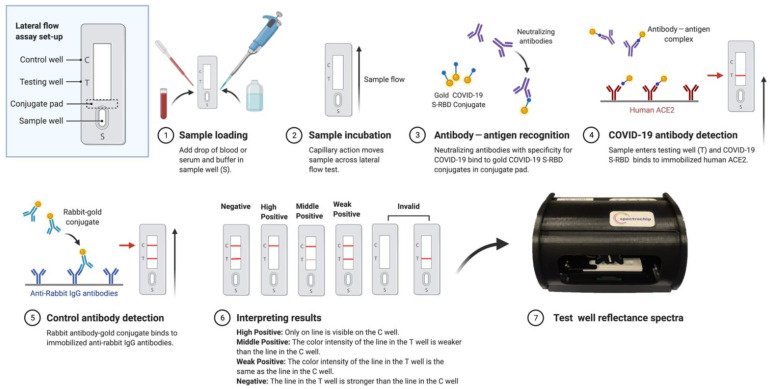
SARS-CoV-2 Neutralizing Antibody Rapid Test Cassette workflow coupled with a spectrum-based reader. This newly quantitative system required 60 µL of serum and 40 µL of dilution buffer to be added to the lateral flow immunoassay and provided results in 10 min. The test cassette was placed inside a spectrum analyzer for the quantitative spectral analysis. This scan took approximately three minutes to complete. Automatic scanning of the rapid test cassette was activated with software. Full-spectrum antibody reflex optical signals were acquired from the spectral optical module to analyze the neutralizing antibody full-spectrum distribution and concentration.

**Figure 2 vaccines-10-00271-f002:**
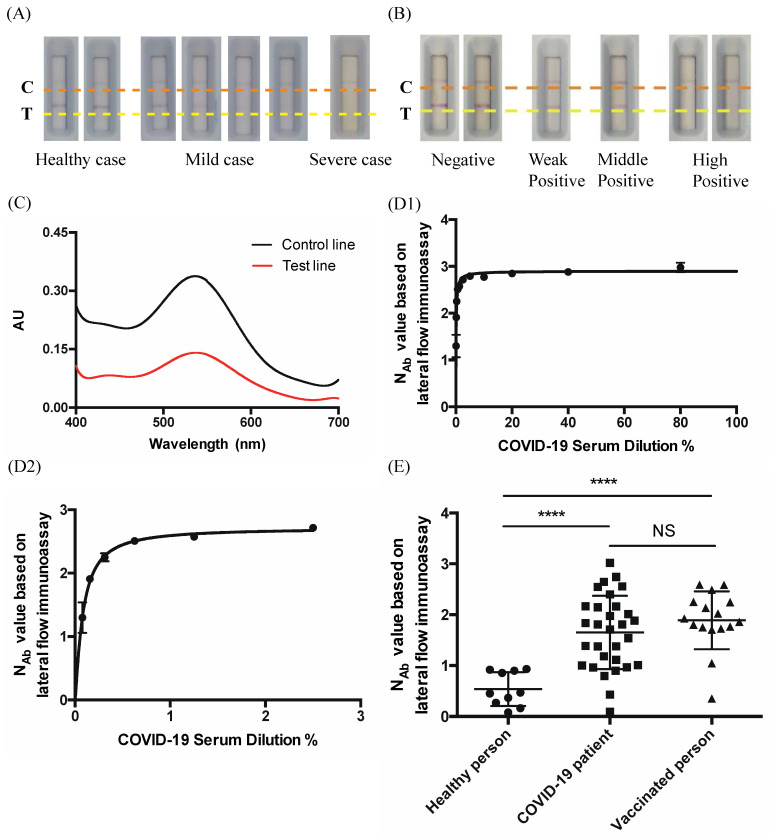
(**A**) SARS-CoV-2 Neutralizing Antibody Rapid Test Cassette (coupled with a spectrum-based reader) loaded with different COVID-19-confirmed cases. Control line region shown as an orange dotted line; test line region shown as a yellow dotted line. (**B**) SARS-CoV-2 Neutralizing Antibody Rapid Test Cassette (coupled with a spectrum-based reader) loaded with samples from different vaccinated individuals. Control line region shown as an orange dotted line; test line region shown as a yellow dotted line. (**C**) The reflectance spectra of a mild case, with the reflectance spectra of the control line region shown as a black solid line; the reflectance spectra of the test line region shown as a red solid line. The *x*-axis is the unit of wavelength (nm). The *y*-axis is the arbitrary unit (AU). (**D1**) Testing the N_Ab_ value based on the lateral flow immunoassay with a dilution series of a severe case patient’s serum. Curve fit by the Hill equation (R-squared = 0.9582), and (**D2**) is part of (**D1**) (R-squared = 0.9649). (**E**) Comparison of the N_Ab_ value based on the lateral flow immunoassay from healthy people (*n* = 10), COVID-19 patients (*n* = 30), and vaccinated people (*n* = 16). **** *p* < 0.0001; NS indicates no significant difference (*p* > 0.05).

**Figure 3 vaccines-10-00271-f003:**
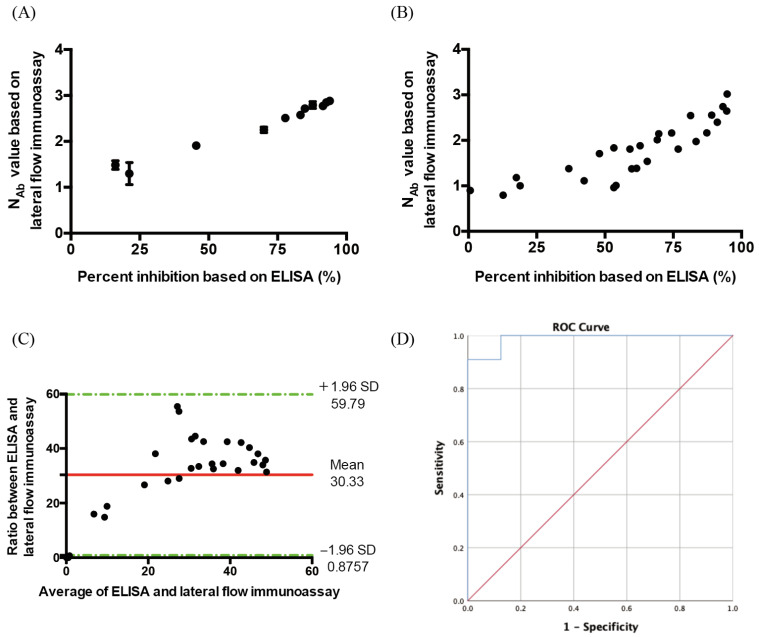
(**A**) Comparison of the lateral flow immunoassay and ELISA for the neutralizing antibody using diluent series serum from a severe case (Rho = 0.9818, *p*-value < 0.0001). The *x*-axis is the unit of percent inhibition based on ELISA. The *y*-axis is the unit of N_Ab_ value based on the lateral flow immunoassay. (**B**) Comparison of the lateral flow immunoassay and ELISA for the neutralizing antibody using serum from patients, including 26 mild cases and 4 severe cases (*n* = 30) (Rho = 0.9288, *p*-value < 0.0001). The *x*-axis is the unit of percent inhibition based on ELISA. The y-axis is the unit of N_Ab_ value based on the lateral flow immunoassay. (**C**) Bland and Altman plot. The ratio between the neutralizing antibody based on the lateral flow immunoassay and ELISA in relation to the mean of the two measurements (*n* = 30). Green lines indicate the limits of agreement (±1.96 standard deviation). (**D**) ROC curve of the neutralizing antibody concentration in COVID-19 cases based on the lateral flow immunoassay and ELISA (*n* = 30) (AUC = 0.989, *p* < 0.001).

**Figure 4 vaccines-10-00271-f004:**
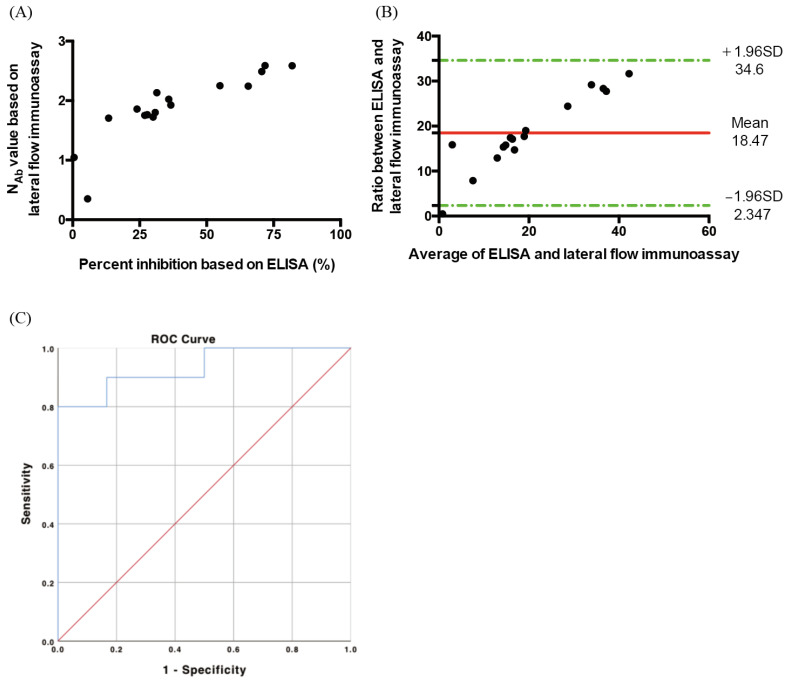
(**A**) Comparison of the lateral flow immunoassay and ELISA for the neutralizing antibody using plasma from vaccinated patients (Rho = 0.9818, *p*-value < 0.0001). The *x*-axis is the unit of percent inhibition based on ELISA. The *y*-axis is the unit of N_Ab_ value based on the lateral flow immunoassay. (**B**) Bland and Altman plot. The ratio between the neutralizing antibody based on the lateral flow immunoassay and ELISA in relation to the mean of the two measurements (*n* = 16). Green lines indicate the limits of agreement (±1.96 standard deviation). (**C**) ROC curve of the neutralizing antibody concentration in COVID-19-vaccinated cases based on the lateral flow immunoassay and ELISA (*n* = 16) (AUC = 0.933, *p* < 0.001).

**Figure 5 vaccines-10-00271-f005:**
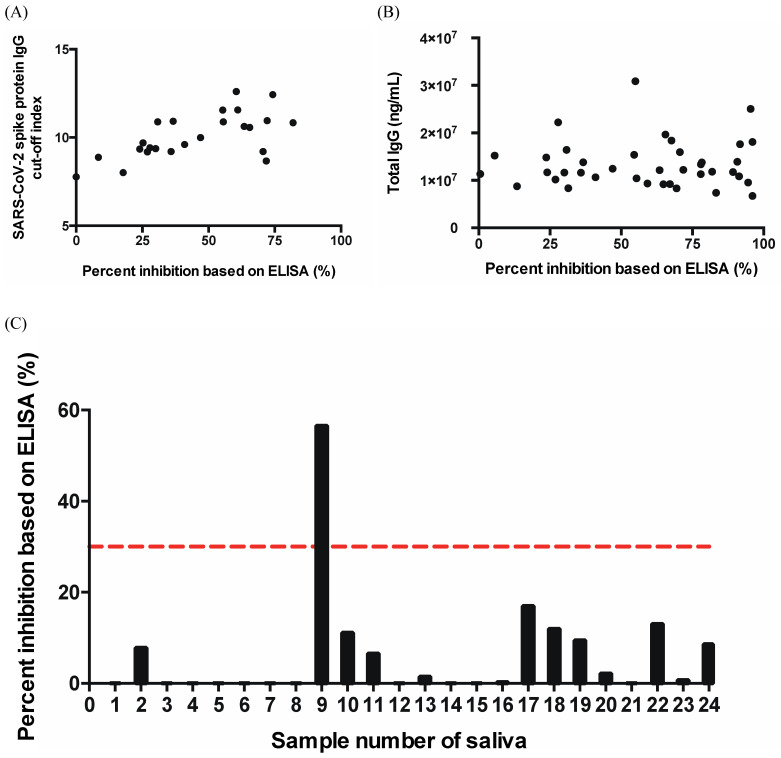
(**A**) Correlation between the SARS-CoV-2 spike protein IgG antibody and neutralizing antibody from ELISA (Rho = 0.5724, *p*-value = 0.0035). The *x*-axis is the percent inhibition of the neutralizing antibody. The *y*-axis is the cut-off index for the SARS-CoV-2 spike protein IgG antibody. (**B**) Correlation between the human total IgG antibody and neutralizing antibody from ELISA. The *x*-axis is the unit of percent inhibition of the neutralizing antibody. The *y*-axis is the amount of total IgG antibody (ng/mL). (**C**) Neutralizing antibody tested from a saliva sample. We labeled the negative percent inhibition as zero. The *x*-axis is the saliva sample number. The *y*-axis is the unit of percent inhibition of the neutralizing antibody. The red dotted line represents the 30% inhibition based on ELISA, which is the cut-off value (positive or negative) of this assay.

**Table 1 vaccines-10-00271-t001:** Results summary of the lateral flow immunoassay and ELISA of enrolled patients and healthy individuals.

Patient	Percent Inhibition Based on ELISA (%)	N_Ab_ ^#^ Value Based on Lateral Flow Immunoassay
1	0.65	0.90
2	0	0.09
3	18.90	1.00
4	42.38	1.11
5	59.15	1.81
6	53.27	1.83
7	59.80	1.38
8	61.59	1.38
9	69.16	2.01
10	65.46	1.54
11	69.69	2.14
12	81.36	2.55
13	83.36	1.98
14	91.16	2.40
15	87.28	2.16
16	0.00	0.96
17	0.00	0.43
18	17.55	1.18
19	0.00	0.97
20	47.95	1.71
21	89.11	2.55
22	62.85	1.88
23	54.01	1.01
24	94.56	2.65
25	74.39	2.16
26	36.76	1.38
27	12.72	0.80
28	76.79	1.81
29	93.21	2.74
30	94.77	3.02
Healthy individual 1	-	0.90
Healthy individual 2	-	0.93
Healthy individual 3	-	0.08
Healthy individual 4	-	0.27
Healthy individual 5	-	0.47
Healthy individual 6	-	0.36
Healthy individual 7	-	0.16
Healthy individual 8	-	0.45
Healthy individual 9	-	0.85
Healthy individual 10	-	0.92

^#^, N_Ab_ means neutralizing antibody.

**Table 2 vaccines-10-00271-t002:** Results summary of the lateral flow immunoassay and ELISA of vaccinated persons.

Patient	Percent Inhibition Based on ELISA (%)	N_Ab_ ^#^ Value Based on Lateral Flow Immunoassay
1	26.91	1.75
2	65.51	2.24
3	70.56	2.49
4	24.03	1.86
5	27.85	1.76
6	30.01	1.72
7	30.81	1.80
8	71.79	2.59
9	81.89	2.59
10	35.86	2.02
11	36.65	1.93
12	5.57	0.35
13	31.40	2.13
14	54.97	2.25
15	0.52	1.05
16	13.43	1.71

^#^, N_Ab_ means neutralizing antibody.

## Data Availability

The datasets of this research are available on request to the corresponding author.

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
