# Peer review of "A Lateral Flow Immunoassay Coupled with a Spectrum-Based Reader for SARS-CoV-2 Neutralizing Antibody Detection"

_vaccines, 2022, doi:10.3390/vaccines10020271_

Round 1

Reviewer 1 Report

This is an interesting article about the study of neutralizing antibodies to COVID-19.

I would like to address a small number of suggestions to you.

General recommendations

Please check all the text for spelling mistakes.

Please correct all references according to journal stile.

References should be described as follows:

Journal Articles:

  1. Author 1; Author 2. Title of the article. Abbreviated Journal Name Year, Volume, page range.

Moreover, you are required to include in-text references in the main body of your work in square brackets [1].

Materials and Methods.

In your results section you present your results about the NAb testing in saliva. Unfortunately, in this manuscript, in materials and methods section information about saliva collection and testing for Nabs was not given.

Please analytically describe when saliva samples were collected from patients, which methods were used for antibodies detection in saliva?

Moreover, please complete your patients' categorization in the material and methods section. It is necessary to describe patients' age in both vaccinated and COVID-19 groups. In addition, the timing from the positive PCR results, when blood samples were taken from COVID-19 patients must be presented.

Please specify, heparin plasma or EDTA plasma samples were collected from vaccinated subjects in this study?

Line 185

Reading O.D. only at 450nm without correction at other nms may be less accurate.

Results

line 199-201 "Blood samples from those with COVID-19 (26 mild cases and 4 severe cases), 10 healthy individuals, and 16 vaccinated patients were used to detect NAb using the lateral flow immunoassay coupled with a spectrum- based reader (Tables 1 & 2)".

You do not describe in materials and methods section including criteria for healthy individuals in this study. Moreover, results about the NAb level in 10 healthy individuals are not presented in Tables 1 and 2.

Why correlation between SARS-CoV-2 spike protein IgG antibody and neutralizing antibodies were presented only for ELISA?

Do you think that detection of neutralizing antibodies in a home setting is clinically necessary? Generally, conclusions are too short. Undeniably, savings in time, cost, and training are significant, but one of the most important factors for one method are sensitivity and specificity.

Reviewer 2 Report

Dear Authors,

thank you very much for your manuscript "A lateral flow immunoassay coupled with a spectrum-based reader for SARS-CoV-2 neutralizing antibody detection"; it was a pleasure to read this well-written manuscript on carrying out a well thought-out study with creative approaches to challenge the test even outside the scope of approval (saliva).

My comments are:

  • please consider not to use the term "global pandemic"; in my opinion, this term is a pleonasm
  • Line 206 right end - "two line are ..." should read "two lines are ..."
  • please provide data on the reproducibility of measurement results obtained by the reader by both using same cassettes and several consecutive cassettes that measured the same sample

Best wishes

Round 2

Reviewer 1 Report

All my corrections and recommendations were added.